# Thrombin Preconditioning Boosts Biogenesis of Extracellular Vesicles from Mesenchymal Stem Cells and Enriches Their Cargo Contents via Protease-Activated Receptor-Mediated Signaling Pathways

**DOI:** 10.3390/ijms20122899

**Published:** 2019-06-14

**Authors:** Dong Kyung Sung, Se In Sung, So Yoon Ahn, Yun Sil Chang, Won Soon Park

**Affiliations:** 1Department of Pediatrics, Samsung Medical Center, Sungkyunkwan University School of Medicine, Seoul 06351, Korea; dbible@skku.edu (D.K.S.); sein.sung@samsung.com (S.I.S.); soyoon.ahn@samsung.com (S.Y.A.); yschang@skku.edu (Y.S.C.); 2Stem Cell and Regenerative Medicine Institute, Samsung Medical Center, Seoul 06351, Korea; 3Department of Health Sciences and Technology, SAIHST, Sungkyunkwan University, Seoul 06351, Korea

**Keywords:** mesenchymal stem cell, extracellular vesicle, thrombin, protease activated receptors

## Abstract

We investigated the role of protease-activated receptor (PAR)-mediated signaling pathways in the biogenesis of human umbilical cord blood-derived mesenchymal stem cell (MSC)-derived extracellular vesicles (EVs) and the enrichment of their cargo content after thrombin preconditioning. Immunoblot analyses showed that MSCs expressed two PAR subtypes: PAR-1 and PAR-3. Thrombin preconditioning significantly accelerated MSC-derived EV biogenesis more than five-fold and enriched their cargo contents by more than two-fold via activation of Rab5, early endosomal antigen (EEA)-1, and the extracellular signal regulated kinase (ERK)1/2 and AKT signaling pathways. Blockage of PAR-1 with the PAR-1-specific antagonist, SCH79797, significantly suppressed the activation of Rab5, EEA-1, and the ERK1/2 and AKT pathways and subsequently increased EV production and enriched EV cargo contents. Combined blockage of PAR-1 and PAR-3 further and significantly inhibited the activation of Rab5, EEA-1, and the ERK1/2 and AKT pathways, accelerated EV production, and enriched EV cargo contents. In summary, thrombin preconditioning boosted the biogenesis of MSC-derived EVs and enriched their cargo contents largely via PAR-1-mediated pathways and partly via PAR-1-independent, PAR-3-mediated activation of Rab5, EEA-1, and the ERK1/2 and AKT signaling pathways.

## 1. Introduction

Extracellular vesicles (EVs), which have a diameter of 40–150 nm, are secretory membrane vesicles released by a variety of cells. EVs contain numerous proteins, lipids, and RNAs that are similar to those present in the originating cells. EVs have been recognized as important messengers for cell-to-cell communication via transfer of the various factors contained therein [1,2,3,4]. Recent studies have shown that the therapeutic efficacy of umbilical cord blood-derived mesenchymal stem cells (UCB-MSCs) in various disorders, such as cardiovascular diseases [5], lung injury [6,7,8], acute kidney injury [9], fetal hypoxic ischemic brain injury [10], skin wound healing [11], and hypoxic pulmonary hypertension [12], is largely mediated by the transfer of mRNAs, miRNAs, and proteins via MSC-derived EVs [6,8,13,14,15,16]. The major advantage of using cell-free MSC-derived EV therapy over transplantation of live MSCs is that EVs can overcome the concerns associated with live cell therapy. Moreover, compared to their parent MSCs, EVs can be stored without losing their biological function, and thus are more suitable for use as an “off the shelf” drug [17]. Despite their promise, only a small amount of EVs is constitutively secreted by MSCs, and the resulting low therapeutic efficacy is a challenge for clinical translation. In our previous study [11], we observed that thrombin preconditioning of MSC-derived EVs accelerated cutaneous wound healing, which was better than that in hypoxia, H_2_O_2_, or lipopolysaccharide exposure, by boosting EV biogenesis and enriching their cargo contents. However, the precise molecular mechanisms underlying the positive effects of thrombin preconditioning on EV biogenesis and their cargo contents have not yet been elucidated.

Thrombin is a serine protease that has a variety of biological activities [18]. Thrombin activates a family of protease activated receptors (PARs) that includes PAR-1, PAR-3, and PAR-4 [19,20]. It has been reported that the neuroprotective effects of thrombin preconditioning are mediated by the activation of PARs [21] and that thrombin induces the secretion of fibronectin by MSCs via PAR-mediated extracellular signal-regulated kinase (ERK) 1/2 activation [22]. In our previous study, we observed that thrombin preconditioning enhanced the proliferation, migration, and tube formation of human umbilical vein endothelial cells (HUVECs) in vitro via phosphorylated (*p*)ERK1/2 and *p*AKT signaling pathways. However, the role of PARs and the precise molecular mechanisms underlying the boosting effects of thrombin preconditioning on the biogenesis of MSC-derived EVs and their cargo contents have not been delineated yet. In the present study, we thus investigated whether thrombin preconditioning promoted the production of MSC-derived EVs and enriches their cargo contents via PAR-mediated activation of Rab5, early endosome antigen (EEA)-1, ERK1/2, and AKT.

## 2. Results

### 2.1. Thrombin Preconditioning Exerts Dose-Dependent Effects on the Biogenesis, Protein Contents, and Characteristics of EVs

Thrombin preconditioning at 1–4 U/mL induced dose-dependent increases in EV production and the levels of cargo proteins such as vascular endothelial growth factor (VEGF), angiogenin, angiopoietin-1, and hepatocyte growth factor (HGF), with peaks observed at 2 and 4 U/mL (Figure 1A,B). Based on these results, an optimal preconditioning dose of 2 U/mL human thrombin was used in subsequent experiments. Early endosomes were labeled with green fluorescence using the CellLight Early Endosomes-green fluorescent protein(GFP) kit; significantly higher green fluorescence, indicative of the number of endosomes, was observed in the thrombin-preconditioned MSCs than in the naive MSCs (Figure 1C,D). In representative SEM and TEM images, more isolated EVs were observed for the thrombin-preconditioned MSCs than in the naive MSCs (Figure 1E). The size of the isolated EVs peaked at 100 nm in diameter for both the thrombin-preconditioned and naive MSCs, although the number of isolated EVs was much higher in the thrombin-preconditioned MSCs than in the naive MSCs (Figure 1F). The EVs isolated from both the thrombin-preconditioned and naive MSCs were positive for the exosome-specific markers CD9, CD63, and CD81, but negative for mitochondrial cytochrome C, GM130, and fibrillarin (Figure 1G).

### 2.2. MSCs Express the Thrombin PARs

Immunoblot analyses were performed to evaluate whether MSCs express the receptors for thrombin. MSCs expressed PAR-1 and PAR-3 but not PAR-2 or PAR-4 (Figure 2A,B), suggesting that thrombin preconditioning might exert its effect on MSCs through PAR-1 and PAR-3. In the control experiment, PAR-3 expression in MSCs was silenced by transfection with PAR-3-specific siRNA but not with scrambled siRNA (Figure 2C).

### 2.3. Thrombin Preconditioning Induces Increases in Early Endosomal Marker Protein Levels and Phosphorylation of the ERK1/2 and AKT Pathways

Thrombin preconditioning significantly increased the levels of early endosome marker proteins such as Rab5 and EEA1, as shown by immunoblotting (Figure 3A,B). Further, thrombin preconditioning activated the ERK1/2 and AKT cascades in UCB-MSCs. Immunoblotting analysis results for phospho(p)ERK1/2, ERK1/2 pAKT, and AKT levels in thrombin-preconditioned MSCs are shown in Figure 3C,D, as compared with their levels in naive MSCs. Thrombin preconditioning significantly increased pERK1/2 and pAKT expression. ERK1/2 and AKT expression levels were increased slightly, but the increase was not statistically significant.

### 2.4. Blockage of PAR Suppresses Thrombin-Induced Rab5 and EEA1 Expression, ERK1/2 and AKT Phosphorylation, as well as EV Production

To investigate how thrombin induced the expression of Rab5 and EEA-1, thrombin-preconditioned MSCs were treated with the PAR-1-specific antagonist SCH79797 [22] and/or PAR-3 was blocked by transfection with a PAR-3-specific siRNA. Thrombin preconditioning-induced increases in Rab5 and EEA-1 levels were more significantly inhibited by SCH79797 than by PAR-3-specific siRNA, but were most significantly inhibited by the combination of SCH79797 and PAR-3-specific siRNA (Figure 3E,F). To determine how thrombin induced the activation of ERK1/2 and AKT, thrombin-preconditioned MSCs were treated with SCH79797 and/or PAR-3 was blocked by PAR-3-specific siRNA. Thrombin preconditioning-induced increases in ERK1/2 and AKT phosphorylation were more significantly inhibited by SCH79797 than by PAR-3-specific siRNA, but were most significantly inhibited by the combination of the two (Figure 3G,H). These results suggest that PAR-1 signaling is more strongly involved, whereas PAR-3 signaling is partially involved, in the thrombin preconditioning-induced activation of Rab5, EEA-1, ERK1/2, and AKT.

To investigate if the genesis of extracellular vesicles is controlled by PAR, thrombin-preconditioned MSCs were treated with the PAR-1-specific antagonist SCH79797 and/or PAR-3 was blocked by transfection with a PAR-3-specific siRNA. To compare the generality of the early endosomes, they were labeled with green fluorescence using the CellLight Early Endosomes-GFP kit after thrombin preconditioning. Increases in the number of early endosomes by thrombin treatment was more significantly inhibited by SCH79797 treatment than by PAR-3 siRNA transfection, and were most significantly inhibited by the combination of SCH79797 treatment and PAR-3 siRNA transfection (Figure 4A,B). We compared the increase in extracellular vesicle number of the UCB-MSC culture medium after treatment with thrombin and thrombin receptor inhibitors. Increases in extracellular vesicle numbers by thrombin treatment were more significantly inhibited by SCH79797 treatment than by PAR-3 siRNA transfection and were most significantly inhibited by the combination of SCH79797 treatment and PAR-3 siRNA transfection (Figure 4C,D). After treatment of thrombin and thrombin receptor inhibitors, we compared the cargo protein enrichment in extracellular vesicles. Increases in cargo protein by thrombin treatment were more significantly inhibited by SCH79797 treatment than by PAR-3 siRNA transfection and were most significantly inhibited by the combination of SCH79797 treatment and PAR-3 siRNA transfection (Figure 4E,F). These results suggest that PAR-1 signaling is strongly involved and PAR-3 signaling is partially involved in thrombin preconditioning-stimulated EV production and cargo protein enrichment.

## 3. Discussion

Recent studies have demonstrated the remarkable therapeutic efficacy of MSC-derived EVs, which are as good as the parental cells, in numerous preclinical disease models [6,7,8,15,16,23]. In addition, EVs alleviate the concerns associated with the use of live cells and do not lose their biological function during storage [17], making cell-free EV therapy more suitable than MSC therapy for “off the shelf” use. However, despite these promising findings, the successful clinical translation of MSC-derived EVs is hampered by the low productivity and limited therapeutic efficacy of naive MSC-derived exosomes. Thrombin preconditioning, which was shown to be non-cytotoxic, altered the characteristics of MSCs, including their surface marker profile, during in vitro adipogenesis and osteogenesis [11,22,24]. In our previous study [11], thrombin preconditioning boosted EV production and increased the levels of various growth factors, including VEGF and angiogenin, in the EVs when compared with other preconditioning regimens, including hypoxia, lipopolysaccharide, and H_2_O_2_. Moreover, thrombin preconditioning enhanced the proliferation, migration, and tube formation of human umbilical venous endothelial cells in vitro via the *p*ERK1/2 and *p*AKT signaling pathways and improved cutaneous wound healing in vivo when compared to other preconditioning regimens. In the present study, thrombin preconditioning increased the production of EVs more than five-fold and increased the cargo protein levels in the EVs more than two-fold when compared with levels in naive MSCs. These findings suggest that thrombin preconditioning could maximize the therapeutic efficacy of MSC-derived EVs by boosting their biogenesis and enriching their cargo contents. Moreover, for biological, physical, or chemical changes that thrombin might have induced in MSCs, thrombin preconditioning was neither cytotoxic nor altered the characteristics of MSCs including their surface marker profiles and in vitro chondrogenesis, adipogenesis, and osteogenesis [11,22,25,26]. Since human thrombin is clinically available, the promising beneficial effects observed in our previous and present studies could be easily translated into clinical practice.

Thrombin is known to be involved in several biological processes, including homeostasis and cytoskeletal reorganization [27], secretion of chemokines, and synthesis of matrix metalloproteinases [22,28,29], through the activation of PAR-1, PAR-3, and PAR-4 [20,27]. In contrast to the study by Chen et al. [22], which revealed PAR-1 and PAR-2 expression in bone marrow-derived MSCs, our study showed that PAR-1 and PAR-3 were expressed in human umbilical cord blood MSC-derived EVs, but PAR-2 or PAR-4 were not. Further studies are needed to confirm this.

The precise molecular mechanisms by which thrombin preconditioning promotes the biogenesis of MSC-derived EVs and their cargo contents have not been elucidated. PAR-mediated signaling via G proteins is known to activate multiple downstream pathways in other cell types [30]. In this study, thrombin preconditioning significantly upregulated the early endosome-specific proteins Rab5 [31] and EEA-1, increased the levels of *p*ERK1/2 [22] and *p*AKT, and promoted the biogenesis of EVs and enriched their cargo contents. These results suggest that thrombin preconditioning enhances the production of MSC-derived EVs and enriches their cargo contents through the activation of Rab5, EEA-1, and the ERK1/2 and AKT pathways.

To further elucidate the role of PARs in thrombin-stimulated EVs biogenesis, MSCs were pretreated with the PAR-1 signaling antagonist SCH79797 [22], and PAR-3 was blocked with siRNA transfection due to the limited number of clinically available PAR-3 antagonists. The activation of Rab5, EEA-1, ERK1/2, and AKT along with the production of EVs by MSCs and their cargo contents were investigated. Our data showed that blockage of PAR-1 significantly suppressed the thrombin-stimulated activation of Rab5 and EEA-1 and phosphorylation of ERK1/2 and AKT as well as the resulting biogenesis of EVs and cargo content enrichment, compared with blockage of PAR-3. Furthermore, combined blockage of PAR-1 and PAR-3 led to a more significant inhibition of thrombin preconditioning-induced activation of Rab5, EEA-1, and the ERK1/2 and AKT pathways, in addition to the resultant increase in EV production and enhancement of cargo contents. These results suggest that while PAR-1 plays a central role, PAR-3 plays an ancillary role as an upstream mediator in this process. Their roles in thrombin preconditioning-induced activation of Rab5, EEA-1, and the ERK1/2 and AKT pathways, and the subsequent increases in EV production and cargo contents are independent of each other.

In summary, thrombin preconditioning of human UCB-derived MSCs promoted the biogenesis of EVs by more than five-fold and enriched their cargo contents by more than two-fold via activation of Rab5, EEA-1, and the ERK1/2 and AKT pathways. PAR-1 largely and PAR-3 partly, but independently, upregulated the activation of the signaling pathways and the resultant increase in EV production and enrichment of EV cargo contents. Given that the successful clinical translation of MSC-derived EVs is hindered by their low productivity, our data showing enhanced biogenesis of EVs by thrombin via PAR-1 and -3 pathways indicate that developing more PAR-1- and -3-specific agonists might enhance EV production and improve the therapeutic efficacy without any side-effects.

## 4. Materials and Methods 

### 4.1. Mesenchymal Stem Cells

Human umbilical cord blood (UCB)-derived MSCs were purchased from Medipost Co., Ltd. (Seoul, Korea) and were used in this study [32,33]; they were obtained from a single donor with maternal informed consent and cultivated and manufactured in strict compliance with good manufacturing practice at passage 6. The cells expressed CD105 (99.6%) and CD73 (96.3%) but not CD34 (0.1%), CD45 (0.2%), or CD14 (0.1%) and were positive for human leukocyte antigen (HLA)-AB (96.8%) but not HLA-DR (0.1%). Characterization of these MSCs [34], their differentiation potential [32,35], immunophenotypic results [35], and karyotype stability up to passage 11 [35] have been previously reported [34].

### 4.2. Preconditioning of MSCs

UCB-MSCs were cultured in α-MEM (Gibco, Grand Island, NY, USA) supplemented with 10% (*v*/*v*) fetal bovine serum(FBS, Gibco), 100 units/mL penicillin, and 100 μg/mL streptomycin (Invitrogen, Carlsbad, CA, USA) under standard culture conditions. At approximately 90% confluence in the culture plate, the cells were washed with phosphate buffered saline(PBS) three times to remove contaminating FBS-derived exosomes. The cells were then incubated with fresh serum-free α-MEM supplemented with human recombinant thrombin (1, 2, and 4 U/mL; Sigma-Aldrich, St. Louis, MO, USA) for 6 h. After collection of conditioned media, the cells were counted using the Luna-FL™ system (Logos Biosystems, Anyang-si, Korea) with about 2 × 10^6^ cells per 100 mm culture dish.

### 4.3. Isolation of EVs

After preconditioning, the conditioned medium was centrifuged at 3000 rpm for 30 min at 4 °C (Eppendorf, Hamburg, Germany) to remove cellular debris, and then at 100,000 rpm for 120 min at 4 °C (Beckman, Brea, CA, USA) to sediment the EVs. This pellet was washed twice, re-suspended in sterile PBS, and then stored at −80 °C until use.

### 4.4. Quantification of EV Production

The distribution of EVs was analyzed by measuring the rate of Brownian motion using a NanoSight (NanoSight NS300; Malvern, Worcestershire, UK), which is equipped with fast video capture and particle-tracking software. The obtained EVs were resuspended in PBS (500 μL, 1 mg/mL total protein), and their size and polydispersity were determined. To quantify EV production by a single cell following collection of the conditioned medium, cells were counted using the LUNA-FL system according to the manufacturer’s protocol. The number of EVs produced by a single cell was calculated by dividing the total number of EVs by the number of cells.

### 4.5. Transmission Electron. Microscopy (TEM)

EVs (5 µL) were fixed with 2% glutaraldehyde, loaded on 200-mesh formvar/carbon-coated electron microscopy grids (Electron Microscopy Sciences, Washington, PA, USA), and incubated for 10 min. They were then washed with filtered distilled water and stained with 2% uranyl acetate in water for 1 min. Transmission electron micrographs were obtained using a Tecnai Spirit G2 transmission electron microscope (FEI, Hillsboro, OR, USA) operating at 120 kV.

### 4.6. Scanning Electron. Microscopy (SEM)

Isolated EVs were fixed in 2.5% glutaraldehyde and loaded on a polycarbonate membrane. The membrane was washed once with PBS and with water and then dehydrated with acetone. The acetone was removed by critical point drying using liquid carbon dioxide. Samples were mounted on aluminum stubs with carbon tape and mounted on an SEM stub. Following sputter-coating with 3–5 nm platinum, the samples were examined using a SEM (Zeiss Auriga Workstation, Oberkochen, Germany).

### 4.7. PAR1-Specific Inhibitor Treatment

The selective PAR1 antagonist SCH 79797 (*N*^3^-cyclopropyl-7-[[4-(1-methylethyl)phenyl]methyl]-7*H*-pyrrolo[3,2-*f*]quinazoline-1,3-diamine dihydrochloride) was obtained from Tocris (Bristol, UK). UCB-MSCs were treated with 1 µM SCH 79797, a selective inhibitor of PAR1, by adding it to the culture medium 1 h before the addition of thrombin.

### 4.8. PAR3 Knockdown 

To knockdown PAR3, MSCs were transfected with a siRNA targeting RAR3 using Lipofectamine^®^ RNAiMAX Transfection Reagent (Invitrogen, Carlsbad, CA, USA) according to the manufacturer’s protocol. MSCs were transfected with a scrambled siRNA, as a negative control, by the same method. Control and PAR3 siRNAs were purchased from Santa Cruz Biotechnology (Santa Cruz, CA, USA).

### 4.9. Early Endosome Labeling

Early endosomes were labeled using Cell Light^®^ Reagent-green fluorescent protein(GFP), BacMam 2.0 (Thermo Fisher Scientific, San Jose, CA, USA) according to the manufacturer’s recommendations. Briefly, MSCs were seeded at a density of 1.5 × 10^4^ cells per well onto 12-well plates in complete growth medium. After the cells attached, BacMam 2.0 reagent was added at a concentration of 40 particles per cell (PPC). Then, Cell Light^®^ Early endosomes-GFP, BacMam 2.0 was used to label early endosomes (to measure Rab5a-GFP expression). To estimate the number of GFP-labeled endosomes, the optical density of the green immunofluorescence was measured using ImageJ (National Institutes of Health, Bethesda, MD, USA).

### 4.10. Bioplex Assay

The cytokine levels in EVs were analyzed by ELISA. A homogenate of isolated EVs was added to a well containing 0.1 mL of the lysis buffer from the ELISA kit. The EV preparations were normalized for protein content, as determined by the Bradford method, and 1 μg of protein was loaded into each well. A Fluorokine^®^ MAP, Human Angiogenesis Custom Premix Kit A (R&D Systems, Minneapolis, MN, USA), was used to quantify angiogenin, angiopoietin-1, VEGF, and HGF in EVs, according to the manufacturer’s instructions.

### 4.11. Immunoblot Analysis

UCB-MSC and EV preparations were lysed by adding an equal volume of RIPA buffer (Sigma-Aldrich, St. Louis, MO, USA). The lysate preparations were normalized for protein content, as determined by the Bradford method. Samples containing 10 μg of protein were mixed with loading buffer containing β-mercaptoethanol, boiled for 10 min, separated in a 12% SDS-polyacrylamide gel gradient (SDS-PAGE), and electrophoretically transferred to nitrocellulose membranes. The membranes were blocked with 5% Bovine serum albumin(BSA) in 1× phosphate buffered saline (PBS-T) containing 0.5% Tween-20 at room temperature (RT) and incubated with the primary antibodies for 1 h at RT. The membranes were washed with 1× PBS-T and then incubated with the secondary antibodies, anti-mouse or anti-rabbit horseradish peroxidase -conjugated immunoglobulin G (1:2000), for 1 h at RT with agitation. Following washing with PBS-T, the protein bands were detected using ECL Select chemiluminescence reagent (GE Healthcare Life Sciences, Piscataway, NJ, USA), and images were acquired with X-ray film.

### 4.12. Statistical Analyses

All quantitative results were obtained from triplicate samples. Data are expressed as the mean ± SD, and statistical analyses were carried out using two-sample *t*-tests to compare two groups and one-way analysis of variance (ANOVA) for three groups. A *p* value less than 0.05 was considered statistically significant.

## Figures and Tables

**Figure 1 ijms-20-02899-f001:**
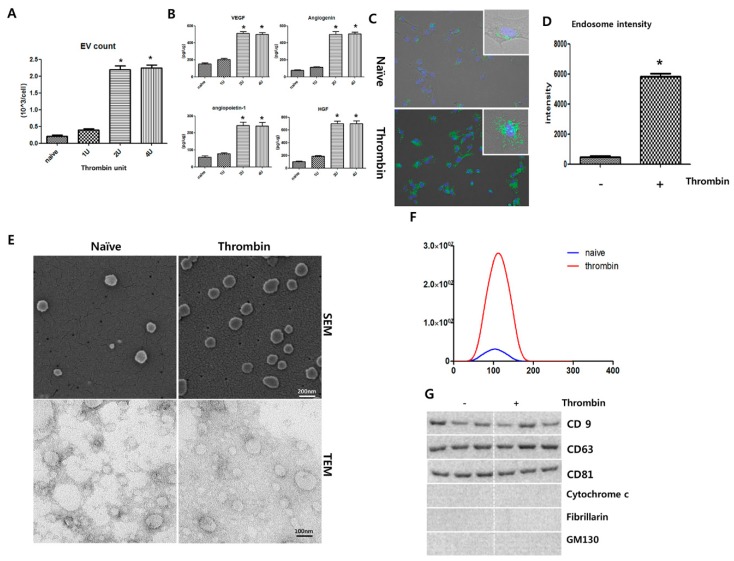
Dose effect of thrombin preconditioning on extracellular vesicle (EV) production and characterization of EVs derived from naive or thrombin-preconditioned umbilical cord-derived mesenchymal stem cells. Extracellular vesicles (EVs) were isolated from the conditioned media of umbilical cord-derived mesenchymal stem cell (UCB-MSC) cultures using ultra-centrifugation. (**A**) Thrombin preconditioning (1 to 4 U) induced a dose-dependent increase in EV production by UCB-MSCs. (**B**) Angiogenin, angiopoietin-1, hepatocyte growth factor (HGF), and vascular endothelial growth factor (VEGF) were measured in the EVs using multiplex ELISA, and the levels of these growth factors increased with the increase in thrombin concentration (*n* = 6 per analysis). MSCs were preconditioned with thrombin (2 U) for 6 h. After thrombin treatment, the number of endosomes produced by the UCB-MSCs was determined by labeling early endosomes with green fluorescent protein(GFP). EVs were isolated from conditioned media of cultures for UCB-MSCs using ultra-centrifugation. (**C**) Endosomes are shown labeled with GFP (green), and the nuclei are shown labeled with 4′,6-Diamidino-2-Phenylindole, Dihydrochloride (DAPI-blue). (**D**) The intensity of the endosome-GFP signal as calculated by Image J. (**E**) Scanning electron micrograph (SEM) of EVs loaded on a polycarbonate membrane. Transmission electron micrograph (TEM) of EVs. EVs on copper grids and stained with uranyl acetate. Representative SEM (upper panel) and TEM (lower panel) of isolated EVs from MSCs. (**F**) The size and number of EVs as measured using a NanoSightNS300 and Nanoparticle Tracking Analysis software. (**G**) Representative immunoblot for the organelle marker proteins in MSC-derived EVs, cytochrome C for mitochondria, fibrillarin for the nucleus, GM130 for the Golgi, and CD81, CD63, and CD9 for exosomes. Data are presented as mean ± SD. An asterisk (*) indicates a significant difference vs. naive UCB-MSCs (*n* = 4 per analysis). Data are presented as mean ± SD. An asterisk (*) indicates a significant difference vs. naive UCB-MSCs (*p* < 0.05, two-sample *t*-tests; *n* = 6 per analysis).

**Figure 2 ijms-20-02899-f002:**
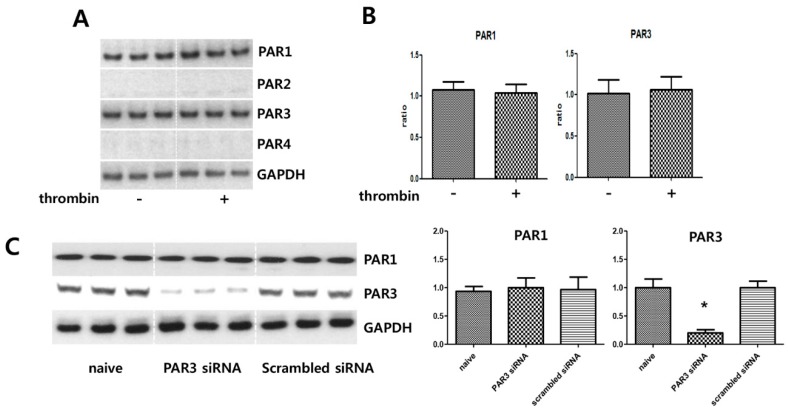
Expression of protease-activated receptors. After thrombin preconditioning (2 U) for 6 h, umbilical cord-derived mesenchymal stem cells (UCB-MSCs) were lysed and subjected to immunoblotting analysis with the indicated antibodies. (**A**) Immunoblot analysis of the expression of protease-activated receptors (PARs) in UCB-MSCs. After thrombin preconditioning, the levels of PARs in the UCB-MSCs were determined by immunoblotting analysis. (**B**) Bar graph showing quantification of the amounts of each PAR. (**C**) Lysates from UCB-MSCs treated with the indicated siRNAs for 24 h were subjected to immunoblot analysis. Cell lysates were analyzed by immunoblotting, and the protein levels were normalized to GAPDH. An asterisk (*) indicates a significant difference vs. naive EVs.

**Figure 3 ijms-20-02899-f003:**
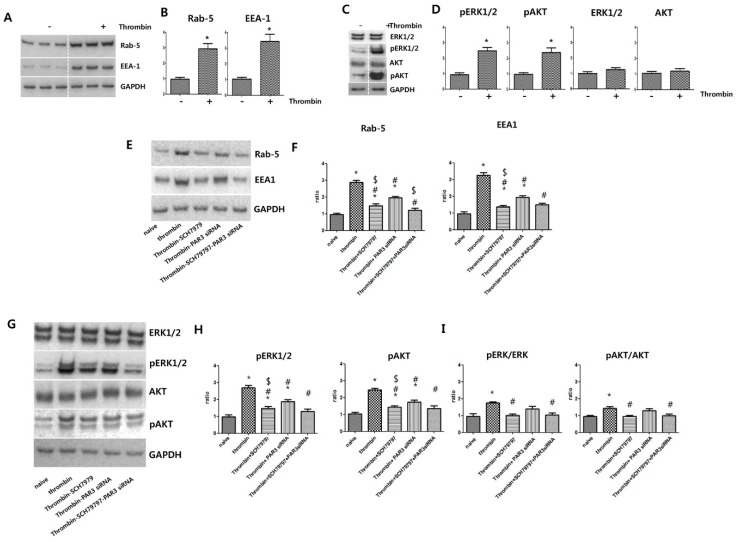
Protease-activated receptors, PAR1 and PAR3, are involved in extracellular vesicle production and phosphorylation of ERK and AKT in umbilical cord-derived mesenchymal stem cells after thrombin preconditioning. Thrombin-preconditioned umbilical cord-derived mesenchymal stem cells (UCB-MSCs) were lysed for immunoblotting analysis with the indicated antibodies. (**A**) After thrombin treatment and inhibition, changes in the protein levels of endosome markers in the UCB-MSCs were assessed by immunoblotting. (**B**) Bar graph showing quantification of Rab-5 and EEA1 levels. (**C**) After thrombin treatment and inhibition, changes in the protein levels of phosphorylated (p)ERK1/2, *p*AKT, ERK1/2, and AKT in the UCB-MSCs were assessed by immunoblotting. (**D**) Bar graph showing quantification of *p*ERK1/2 *p*AKT, ERK1/2, and AKT. (**E**) After thrombin treatment and inhibition, changes in the levels of endosome markers in the UCB-MSCs were determined by immunoblotting. (**F**) Bar graph showing quantification of Rab-5 and EEA1 levels. (**G**) After thrombin treatment, changes in the expression of *p*ERK1/2 and *p*AKT in the UCB-MSCs were determined by immunoblotting. (**H**) Bar graph showing quantification of *p*ERK1/2 and *p*AKT levels. (**I**) Bar graph showing quantification of the ratio of *p*ERK/ERK and *p*AKT/AKT. Data are presented as mean ± SD. An asterisk (*) indicates a significant difference vs. naive EVs, a number sign (#) indicates a significant difference vs. thrombin-treated UCB-MSC and, a dollar sign ($) indicates a significant difference vs. thrombin + PAR3 siRNA UCB-MSCs (*p* < 0.05, two-sample *t*-test; *n* = 5 per analysis).

**Figure 4 ijms-20-02899-f004:**
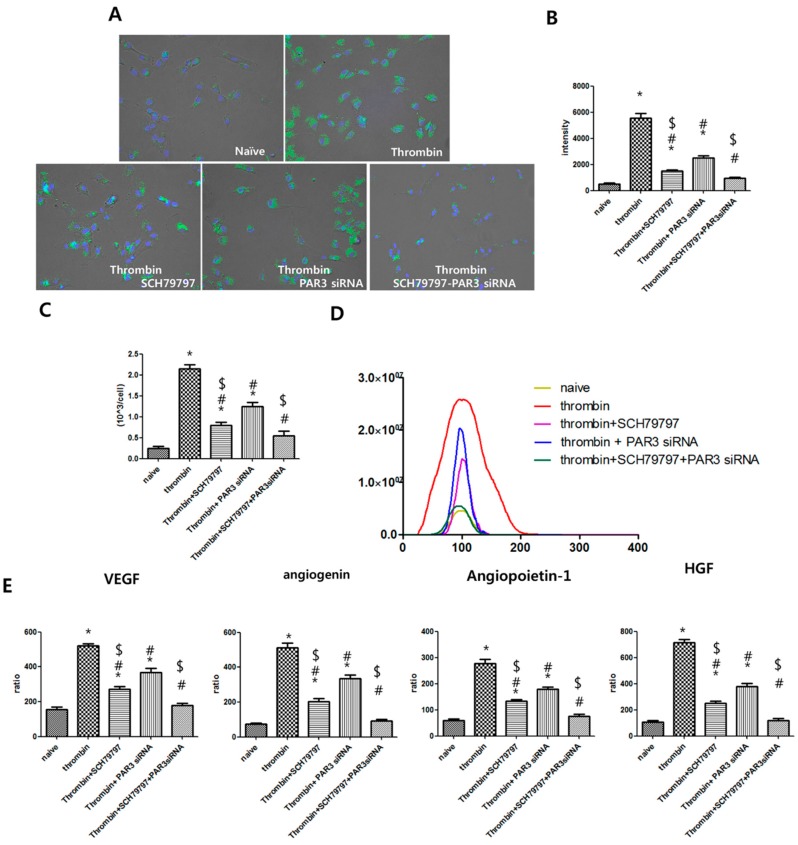
Protease-activated receptors, PAR1 and PAR3, are involved in endosome production by umbilical cord-derived mesenchymal stem cells and protein cargo contents in extracellular vesicles after thrombin preconditioning. (**A**) After thrombin preconditioning of umbilical cord-derived mesenchymal stem cells (UCB-MSCs), the number of endosomes was determined by labeling early endosomes with green fluorescent protein(GFP-green) and the nuclei with 4′,6-Diamidino-2-Phenylindole, Dihydrochloride (DAPI-blue). (**B**) Bar graph showing the quantified intensities of the endosome-GFP signal. (**C**) After treatment with thrombin and thrombin inhibitor, the number of extracellular vesicles (EVs) was determined using a NanoSightNS300. (**D**) The size and number distribution of EVs as measured and analyzed by Nanoparticle Tracking Analysis software. (**E**) After treatment with thrombin and thrombin inhibitor, the levels of VEGF, angiogenin, angiopoietin, and HGF in the EVs were measured by multiplex ELISA. Following treatment with thrombin and thrombin inhibition, EVs were isolated from the conditioned media of UCB-MSC cultures, and VEGF, angiogenin, angiopoietin, and HGF protein levels were assessed. An asterisk (*) indicates a significant difference vs. naive EVs, a number sign (#) indicates a significant difference vs. thrombin-treated UCB-MSC and a dollar sign ($) indicates a significant difference vs. thrombin + PAR3 siRNA UCB-MSC (*p* < 0.05, two-sample *t*-test; *n* = 6 per analysis).

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
