# Peer review of "Thrombin Preconditioning Boosts Biogenesis of Extracellular Vesicles from Mesenchymal Stem Cells and Enriches Their Cargo Contents via Protease-Activated Receptor-Mediated Signaling Pathways"

_ijms, 2019, doi:10.3390/ijms20122899_

Round 1

Reviewer 1 Report

The manuscript entitled “Thrombin preconditioning boosts biogenesis of extracellular vesicles from mesenchymal stem cells and enriches their cargo contents via protease-activated receptor-mediated signaling pathways” describes the pathway activated by thrombin to increase the production of extracellular vesicles (EV) by mesenchymal stem cells. It is the mechanistic explanation of the previous paper of the authors (Sung et al. J Clin Med 2019). The message is clear, however I have some major concerns that need to be addressed in order to consider the paper for publication.

1-    I think that the “results” section needs to be re-organized to be more clear. In my opinion, the first paragraph of the section should describe the effects of thrombin treatment in EV induction; thus, the results of figure 3 and the figure 3 itself should be included in paragraph 3.1 and in figure 1, respectively. As a consequence, the first paragraph will describe the dose-dependent effect of thrombin, demonstrated by the EV count, the cytokine content, the GFP staining, the SEM and TEM images and by the characterization of the physical features of the vesicles.

2-    The results of figure 2 should be included in another paragraph dedicated to the mechanism activated by thrombin to induce EV production.

3-    Paragraph 3.4 and figure 4 should then become paragraph 3.3 and figure 3, respectively. Figure 5 will become figure 4.

There are some other issues regarding the results:

1-    Figure 1. EV production is strongly induced after treatment with 2U and 4U of thrombin (»2200 and »2300 EV/cell, respectively). In their previous work (Sung et al. J Clin Med 2019), after 5U of thrombin, the EV were 500/cells. Could the authors explain these great differences?

2-    Figure 2. Why in WB 2A the authors used Tubulin as housekeeper and then shift to GAPDH in other WBs?

3-    Figure 2 F-G and figure 4 E-F. When proteins’ phosphorylation is analyzed, it should be also measured the content of the total proteins (i.e. total ERK1/2 and total AKT) and then evaluated the ratio between the phosphorylated form and the total protein in order to exclude that the increase in phosphorylation could be due to an increase also in protein expression. Did the authors evaluate the content of total ERK1/2 and total AKT?

4-    Figure 2. All the histograms should have the same dimensions.

5-    Figure 4E. The WB image should include the name of the samples in the different lanes.

6-    Figure 4E-4F. I have some doubts in pAKT quantification: from the WB image reported, it is not possible that bar 3 (thrombin+SCH79797) and bar 5 (thrombin+SCH79797+PAR3siRNA) have almost the same height.

7-    Figure 4B, figure 4F and figure 5E. The indication of significant differences is confused and redundant: i.e. in Rab-5 histogram in figure 4B, if # is added on bars 3, 4 and 5, then ‡, $, and @ are redundant on bar 2. Please check all the redundant marks.

In “Materials and methods” section, paragraph 2.10 describes the extraction procedure of wound tissue from rats and the procedure for TNF and IL-6 dosage, but no results were reported, regarding these methods. Please explain this discrepancy.

I think that there are some minor issues, regarding typos, references and lack of information that need to be addressed.

Lane 37. Cardiovascular diseaseS.

Lane 37. Ref 6 should be substituted with Ghiroldi et al. (PMID 30332812) that includes a paragraph of umbilical cord blood-derived MSC.

Lane 37. Ref 10 should be substituted with Yun et al. (PMID 30939749) that describes the use of MSCs and not only of microvesicles. Instead, ref 10 should be maintained in lane 39.

Lane 38. Ref 11 should be substituted with Alencar et al. that is more recent and describes the use of MSCs (PMID 30574088).

Lane 39. Ref 11 should be added.

Lane 73. “At approximately 90% confluence” has a different font dimension. Please also add the plate or the number of cells used for the experiments.

Lane 79. “for 120 min AT 4°C”.

Lane 105. The time of SCH79797 treatment is not reported. Please add.

Lane 107. “with A siRNA”.

Lane 226-228. Please reformulate the sentence in a clearer form; maybe it should be divided into two different sentences.

Lane 223-225 and Lane 229-230. The sentences are almost the same. Please rewrite.

Lane 315. I think that a final sentence, highlighting the importance of the findings described and the possible implication for the stem cell therapy should be added.

Author Response

Comments and Suggestions for Authors

The manuscript entitled “Thrombin preconditioning boosts biogenesis of extracellular vesicles from mesenchymal stem cells and enriches their cargo contents via protease-activated receptor-mediated signaling pathways” describes the pathway activated by thrombin to increase the production of extracellular vesicles (EV) by mesenchymal stem cells. It is the mechanistic explanation of the previous paper of the authors (Sung et al. J Clin Med 2019). The message is clear, however I have some major concerns that need to be addressed in order to consider the paper for publication.

àWe thank the reviewer for the careful review of our manuscript and the kind comments on our work. We have corrected and improved our manuscript, particularly the figure legends and graphs, in strict accordance with the reviewer’s recommendations.

I think that the “results” section needs to be re-organized to be more clear. In my opinion, the first paragraph of the section should describe the effects of thrombin treatment in EV induction; thus, the results of figure 3 and the figure 3 itself should be included in paragraph 3.1 and in figure 1, respectively. As a consequence, the first paragraph will describe the dose-dependent effect of thrombin, demonstrated by the EV count, the cytokine content, the GFP staining, the SEM and TEM images and by the characterization of the physical features of the vesicles.

à We thank the reviewer for the helpful comments. In keeping with the reviewer’s recommendation, we have re-organized the “Results” section for clarity, and described the dose-dependent effect of thrombin, demonstrated by the EV count, the cytokine content, the GFP staining, the SEM and TEM images and by the characterization of the physical features of the vesicles, in the first paragraph of the Results section of our revised manuscript. (Figure 1 and lines 65~78)

The results of figure 2 should be included in another paragraph dedicated to the mechanism activated by thrombin to induce EV production.

Based on the reviewer’s recommendation, the results of Fig. 2C are described in another paragraph, dedicated to the mechanism by which thrombin induces EV production, in the “Results” section of our revised manuscript (Figure 2, 3 and lines 104~146).

Paragraph 3.4 and figure 4 should then become paragraph 3.3 and figure 3, respectively. Figure 5 will become figure 4.

We have restructured the paragraphs and figures in our revised manuscript in strict accordance with the reviewer’s recommendation (Figure 3, 4 and paragraph 3.3, lines 121~181).

There are some other issues regarding the results:

Figure 1. EV production is strongly induced after treatment with 2U and 4U of thrombin (»2200 and »2300 EV/cell, respectively). In their previous work (Sung et al. J Clin Med 2019), after 5U of thrombin, the EV were 500/cells. Could the authors explain these great differences?

We apologize for the lack of adequate explanation of the differences in thrombin used in our previous and present studies. In the present study, we used human recombinant thrombin instead of the bovine thrombin used in our previous study (Sung et al. J Clin Med 2019), as per the recommendation of the Korean FDA for successful clinical translation in the near future. Therefore, the differences in dose responses of thrombin obtained from our previous and present studies might be attributable to the differences between bovine and human thrombin, indicating that human thrombin is more pure and potent than bovine thrombin. This description has been incorporated into the Methods section (lines 276~278) of the revised manuscript.

Figure 2. Why in WB 2A the authors used Tubulin as housekeeper and then shift to GAPDH in other WBs?

The reason for using tubulin as the housekeeping protein in WB 2A, but using GAPDH in other WBs was just a matter of differing time periods when the experiments were conducted- they were partly conducted last year and partly this year. We were using tubulin, rather than GAPDH as a housekeeping control last year. To avoid causing confusion by using two different housekeeping proteins, we have re-done the experiment shown in WB 2A using GAPDH, and have incorporated these results in our revised manuscript (Figure 2).

Figure 2 F-G and figure 4 E-F. When proteins’ phosphorylation is analyzed, it should be also measured the content of the total proteins (i.e. total ERK1/2 and total AKT) and then evaluated the ratio between the phosphorylated form and the total protein in order to exclude that the increase in phosphorylation could be due to an increase also in protein expression. Did the authors evaluate the content of total ERK1/2 and total AKT?

 As per the reviewer’s recommendation, we measured the content of the total proteins (i.e. total ERK1/2 and total AKT) and then evaluated the ratio between the phosphorylated form and the total protein in order to exclude the possibility that the increase in phosphorylation observed was due to an increase in protein expression. We incorporated these findings into Figure 3 of our revised manuscript. (Figure 3 and line 121~162)

Figure 2. All the histograms should have the same dimensions.

As per the reviewer’s recommendation, all histograms have been modified to have the same dimensions. (Figure 3)

Figure 4E. The WB image should include the name of the samples in the different lanes.

We apologize for this mistake. We have added the names of the samples in the different lanes (Figure 3)

 Figure 4E-4F. I have some doubts in pAKT quantification: from the WB image reported, it is not possible that bar 3 (thrombin+SCH79797) and bar 5 (thrombin+SCH79797+PAR3siRNA) have almost the same height.

We thank the reviewer for pointing out the error in Figure 4E, we have corrected the figure (Figure 4E).

Figure 4B, figure 4F and figure 5E. The indication of significant differences is confused and redundant: i.e. in Rab-5 histogram in figure 4B, if # is added on bars 3, 4 and 5, then ‡, $, and @ are redundant on bar 2. Please check all the redundant marks.

à As per the reviewer’s recommendation, we checked and deleted all redundant marks (Figure 3, 4).

In “Materials and methods” section, paragraph 2.10 describes the extraction procedure of wound tissue from rats and the procedure for TNF and IL-6 dosage, but no results were reported, regarding these methods. Please explain this discrepancy.

 We apologize for this mistake. We have revised the multiplex ELISA method (lines 334~337).

I think that there are some minor issues, regarding typos, references and lack of information that need to be addressed.

Lane 37. Cardiovascular diseaseS.

We have corrected this typographical error in the revised manuscript (line 37).

 Lane 37. Ref 6 should be substituted with Ghiroldi et al. (PMID 30332812) that includes a paragraph of umbilical cord blood-derived MSC.

As per the reviewer’s recommendation, we have substituted Ref. 6 with Ghiroldi et al. (PMID 30332812), and included a paragraph on umbilical cord blood-derived MSCs in our revised manuscript (line 37).

Lane 37. Ref 10 should be substituted with Yun et al. (PMID 30939749) that describes the use of MSCs and not only of microvesicles. Instead, ref 10 should be maintained in lane 39.

As per the reviewer’s recommendation, we have substituted Ref. 10 with Yun et al. (PMID 30939749), and modified the revised manuscript appropriately (line 48). 

Lane 38. Ref 11 should be substituted with Alencar et al. that is more recent and describes the use of MSCs (PMID 30574088).

àAs per the reviewer’s recommendation, we have substituted Ref. 11 with Alencar et al. (PMID 30574088), which is more recent and describes the use of MSCs in our revised manuscript (line 37).

Lane 39. Ref 11 should be added.

àAs per the reviewer’s recommendation, we have added Ref 11 in line 40.

Lane 73. “At approximately 90% confluence” has a different font dimension. Please also add the plate or the number of cells used for the experiments.

àAs per the reviewer’s recommendation, we have corrected the font size in line 274, and also added information on the plate or the number of cells used for the experiments in our revised manuscript (lines 277~279).  

Lane 79. “for 120 min AT 4°C”.

àWe have corrected the typographical error (line 283).

Lane 105. The time of SCH79797 treatment is not reported. Please add.

 àWe have added the time of SCH79797 treatment in our revised manuscript (lines 314~315).

Lane 107. “with A siRNA”.

à We have corrected the typographical error (line 318).

Lane 226-228. Please reformulate the sentence in a clearer form; maybe it should be divided into two different sentences

àAs per the reviewer’s recommendation, we have divided lines 226-228 into two different sentences in our revised manuscript, to improve clarity (lines 164~180).

Lane 223-225 and Lane 229-230. The sentences are almost the same. Please rewrite.

àAs per the reviewer’s recommendation, we have rewritten lines 223-225 and lines 229-230 in our revised manuscript (lines 121~179)

Lane 315. I think that a final sentence, highlighting the importance of the findings described and the possible implication for the stem cell therapy should be added.

 àAs per the reviewer’s recommendation, we have added a final sentence highlighting the importance of the findings described and the possible implications for stem cell therapy in our revised manuscript (lines 256~259).

Reviewer 2 Report

The manuscript by Sung and co-authors describes the effect of the thrombin preconditioning on production and secretion of extracellular vesicles by human umbilical cord blood Mesenchymal Stem Cells. The work is interesting and data presented are sufficient to support the aim. However, the manuscript needs to be improved in all sections
Major points:
-The authors have to clarify the rationale of the study in the instruction.
-The authors have to improve the description of the results. In particular, they have to explain better the experiments made with the MSCs and those with EVs. Moreover, the comparison of MSCs vs EVs data might help the readers
-The authors have to improve the quality of the figures.
-Authors have to improve the discussion. In particular, they have to explain the adverse effect of thrombin preconditioning in human umbilical cord blood Mesenchymal Stem Cells.

Author Response

Open Review 2

Comments and Suggestions for Authors

The manuscript by Sung and co-authors describes the effect of the thrombin preconditioning on production and secretion of extracellular vesicles by human umbilical cord blood Mesenchymal Stem Cells. The work is interesting and data presented are sufficient to support the aim. However, the manuscript needs to be improved in all sections

àWe thank the reviewer for the careful review of our manuscript and the kind comments on our work. We have corrected and improved our manuscript, particularly the figure legends and graphs, in strict accordance with the reviewer’s recommendations.

Major points:
1. The authors have to clarify the rationale of the study in the instruction.

àWe thank the reviewer for the helpful comments, and we have clarified the rationale for this study in the “Introduction” section of the revised manuscript. (lines 57~59).

2. The authors have to improve the description of the results. In particular, they have to explain better the experiments made with the MSCs and those with EVs. Moreover, the comparison of MSCs vs EVs data might help the readers

àWe thank the reviewer for the insightful comments. We tried to better explain the experiments with the MSCs, including thrombin dose response to EV production, PAR expression, response to PAR-1 antagonist, and PAR-3 siRNA transfection, and subsequent Rab5, EEA-1, pERK1/2, and pAKT pathways, and those with EVs, including cargo content measurements of EVs, in our revised manuscript. However, comparison of MSC and EV data was not possible due to the experimental setting of this study. 

3. The authors have to improve the quality of the figures.

As per the reviewer’s recommendation, we have improved the quality of the figures.

4. Authors have to improve the discussion. In particular, they have to explain the adverse effect of thrombin preconditioning in human umbilical cord blood Mesenchymal Stem Cells.

àWe thank the reviewer for the helpful comments. We have added the potential adverse effects that thrombin preconditioning might have induced in MSCs in the Discussion section of the revised manuscript (lines 218~221).

Round 2

Reviewer 1 Report

The revised form of the article "Thrombin preconditioning boosts biogenesis of extracellular vesicles from mesenchymal stem cells and enriches their cargo contents via protease-activated receptor-mediated signaling pathways" is now more clear and all the concerns previously highlighted have been completely addressed.

I suggest the authors to carefully check the manuscript for typos or typing errors (double spaces, no spaces, etc.)

Reviewer 2 Report

no comments